# Application of the IPDfromKM-Shiny Method to Compare the Efficacy of Novel Treatments Aimed at the Same Disease Condition: A Report of 14 Analyses

**DOI:** 10.3390/cancers15061633

**Published:** 2023-03-07

**Authors:** Andrea Messori, Vera Damuzzo, Melania Rivano, Luca Cancanelli, Lorenzo Di Spazio, Andrea Ossato, Marco Chiumente, Daniele Mengato

**Affiliations:** 1Unità di HTA, Regione Toscana, 50139 Firenze, Italy; 2Dipartimento Politiche del Farmaco, Azienda ULSS 2 Marca Trevigiana, 31100 Treviso, Italy; 3Farmacia Ospedaliera, Ospedale Binaghi, 09121 Cagliari, Italy; 4Farmacia Ospedaliera Nord, Ospedale Santa Chiara di Trento, Azienda Provinciale per i Servizi Sanitari (APSS), 38122 Trento, Italy; 5Scuola di Specializzazione in Farmacia Ospedaliera, Università di Padova, 35128 Padova, Italy; 6Società Italiana di Farmacia Clinica e Terapia (SIFaCT), 10123 Torino, Italy; 7Farmacia Ospedaliera, Azienda Ospedale Università di Padova, 35128 Padova, Italy

**Keywords:** IPDfromKM-Shiny method, indirect comparisons, overall survival, progression free survival, patient data reconstruction

## Abstract

**Simple Summary:**

The clinical development of new cancer drugs is based on clinical trials that compare the new drug with the one representing the standard of care. However, it takes some time for the new drug to become available for patient use. In the meantime, the standard of care may have been updated and therefore the efficacy of the new drug needs to be tested against the new standard of care. This objective is usually pursued by performing indirect comparisons: since no “real” trial is available comparing the new drug with the new standard of care, this comparison is carried out by means of a simulated trial wherein patients of real trials are reconstructed individually, pooled together according to an original design of comparison, and finally subjected to standard statistics. Previously published trials are used to perform this comparison. We describe a new method to perform these indirect comparisons, called the IPDfromKM-Shiny method, that lies midway between simple interpolation and advanced statistics. More than 10 investigations have already been conducted in which this method has been used to perform indirect comparisons, especially in the area of haemato-oncology. In the light of these preliminary experiences, the IPDfromKM-Shiny method is proving to be a valid advancement for conducting comparative research in the area of evidence-based medicine.

**Abstract:**

In the area of evidence-based medicine, the IPDfromKM-Shiny method is an innovative method of survival analysis, midway between artificial intelligence and advanced statistics. Its main characteristic is that an original software investigates the Kaplan-Meier graphs of trials so that individual-patient data are reconstructed. These reconstructed patients represent a new form of original clinical material. The typical objective of investigations based on this method is to analyze the available evidence, especially in oncology, to perform indirect comparisons, and determine the place in therapy of individual agents. This review examined the most recent applications of the IPDfromKM-Shiny method, in which a new web-based software—published in 2021—was used. Reported here are 14 analyses, mostly focused on oncological treatments. Indirect comparisons were based on overall survival or progression free survival. Each of these analyses provided original information to compare treatments with one another and select the most appropriate depending on patient characteristics. These analyses can also be useful to assess equivalence from a regulatory viewpoint. All investigations stressed the importance of heterogeneity to better interpret the evidence generated by IPDfromKM-Shiny investigations. In conclusion, these investigations showed that the reconstruction of individual patient data through this online tool is a promising new method for analyzing trials based on survival endpoints. This new approach deserves further investigation, particularly in the area of indirect comparisons.

## 1. Introduction

In July 2021, three statisticians of the MD Anderson Cancer Center in Houston published a new method [1] that studies a Kaplan-Meier graph and reconstructs individual patient data. This method, which is called IPDfromKM-Shiny Method (abbreviated Shiny Method), has been made available in two versions: the first was developed under the R-platform while the second is a web-based application which is freely available online. The Shiny method is not an entirely new technique because, nearly 10 years ago, the method was available in a few commercial statistical packages (e.g., STATA). The approach of the Shiny method is midway between artificial intelligence and advanced statistics. Prerequisites for reconstructing a patient’s archive are satisfied in most cases because the software only requires the availability of a Kaplan-Meier (KM) graph along with the knowledge of the total number of enrolled patients and the number of events (progressions for progression-free survival curves or deaths for overall survival curves). Of course, the Shiny method can also handle other kinds of time-to-event endpoints that have been represented as a KM graph.

The main applications of the Shiny method in the field of evidence-based medicine (EBM) have been reported beginning from 2022. Indirect comparisons of efficacy across innovative treatments represent the typical investigation of researchers who employ this approach. The Shiny method, in fact, allows simulated comparative trials to be easily performed to evaluate treatments for which no real head-to-head trial has been conducted.

The development of a new drug in haemato-oncology aims at the demonstration of superiority, in terms of overall survival (OS) or progression-free survival (PFS), of the new therapeutic option versus the standard of care (SOC) [1,2]. When the trial is based on a time-to-event endpoint, KM curves are the typical statistical tool to report and interpret the results of these analyses. It is well-known that, in recent years, more and more therapeutic regimens sharing the same clinical indication are being investigated, particularly in oncology, and many of these have been ultimately approved. This raises the need to compare the efficacy of newly developed treatments with the corresponding SOC to determine the place in therapy for every new agent. In this framework, since phase-II studies of newly developed agents are more frequent than randomized phase-III studies, indirect cross trial comparisons are increasingly being required owing to the lack of “real” trials. In this framework, the Shiny method [3,4] has emerged in the last two years as a simple but powerful tool to perform these indirect comparisons based on survival endpoints, such as OS and PFS. In fact, in analyzing the graphs of KM curves, the Shiny method allows the reconstruction of individual patient data and, in this way, generates the patients’ database of the clinical trial (denoted as “reconstructed patient-level data” or “reconstructed individual patient data”). After generating the databases of reconstructed patients, appropriate cross trial comparisons are designed and simulated comparative trials can be conducted. The final product of these analyses is represented by a multi-trial survival graph that includes the KM curves of all treatments under examination. These simulated patients are analyzed through standard survival statistics; comparisons between patient groups receiving different treatments are handled through standard statistical parameters, e.g., the hazard ratio (HR).

The aim of this paper is to describe the application of the Shiny method for comparative assessment of the efficacy of novel treatments, with special emphasis on haemato-oncological agents.

## 2. Indirect Comparison of the Efficacy among Treatments Aimed at the Same Disease Condition: Description of 14 Experiences 

This series of 14 experiences of application of the Shiny method was identified through a standard PubMed search (date of last search: 14 January 2023; keywords: <<(survival OR Kaplan-Meier) AND (IPDfromKM OR Shiny)>>) combined with manual Google queries. Eligible papers were 133 from PubMed and 90 from other sources. After examination of the full text of these articles, 14 were found to report a Shiny method analysis. Figure 1 shows the PRISMA flowchart of this selection process. These 14 papers are described and commented in this section of the present paper. 

The IPDfromKM software is freely available on the Internet and does not require any login to run. There is also an R-code version that can be requested from the authors (Liu, N.; Zhou, Y.; Lee, J.J.) at the University of Houston (see Ref. [1]). It should be noted that IPDfromKM is the correct term to indicate the method described in this paper and that “Shiny” is a more general term that includes other statistical methods above and beyond the IPDfromKM method. However, as a matter of fact, the term Shiny has entered into common use to indicate the IPDfromKM, and therefore this term is used herein. In all the analyses reported in the present article, the method was utilized under the following two conditions: (a) no curves were analyzed if the total number of events was not reported; (b) the information of the number of patients at risk by time intervals (which may be unavailable in several curves or, when available, may suffer from lack of standardization) was never entered into the IPDfromKM software. This operational decision was made to improve the reproducibility of our analyses.

### 2.1. Lutetium in Pre-Treated Metastatic Castration-Resistant Prostate Cancer

This analysis [3], which represented one of the first investigations conducted by our group, studied the efficacy of lutetium in pre-treated metastatic castration-resistant prostate cancer and was mainly aimed at evaluating the performance of the Shiny method under its various aspects. In more detail, based on the endpoint of PFS, this study tested the ability of the Shiny method to detect differences in efficacy when the same treatment is compared between two trials sharing the same design. The results (Figure 2) confirmed that the Shiny method is reliable in pursuing this task. This study focused more on the IPDfromKM tool than on the clinical question related to the use of lutetium in prostate cancer.

### 2.2. First Line Treatment of Chronic Lymphocytic Leukemia (CLL)

This analysis, published in October 2022 [4], represents another typical application of the Shiny method. Over the past decades, numerous first-line treatments for CLL have been proposed as an advancement compared with chlorambucil monotherapy. The Shiny method graph (reported in Figure 3) includes seven treatments; PFS is the endpoint. Most innovations are represented by monoclonal antibodies given as combination treatment; in addition to these antibodies, venetoclax and ibrutinib are of great interest because they open the door to chemo-free regimens. As this example shows, the final Shiny method graph (Figure 3) effectively summarizes the speed at which innovation has proceeded and, more importantly, conveys information on the magnitude of the cumulative overall benefit determined by innovation. CLL is an example in which incremental advancements have been numerous. While no breakthrough innovation has occurred, the impact of these numerous incremental advancements has been as important as that of a single breakthrough innovation. In Figure 3, the final Shiny method graph represents efficacy based on an absolute endpoint reported over a long follow-up and in this way offers an original synthesis on how and to what extent evidence has improved. In this case, the currently best treatment (i.e., ibrutinib plus rituximab/obinutuzumab) ensures a rate of PFS around 75% at 5 years, which is more than 7-fold compared with the PFS around 10% determined by chlorambucil, the undisputed standard of care of some decades ago.

### 2.3. Relapsed or Refractory Multiple Myeloma: Analysis of Pharmacological Treatments and Cell-Based Therapies

Relapsed-refractory multiple myeloma (RRMM) is a hematologic malignancy in which several therapeutic innovations have become available in recent years. Two Shiny method analyses [5,6] focused on the main pharmacological agents (n = 4) and cell-based therapies (n = 2), respectively, while the third [5] dealt only with CAR T-cell products (n = 2) (Figure 5). The analysis by Cancanelli et al. [5] was based on OS (Figure 4A,B), while that by Messori et al. [6] was based on PFS (Figure 5). In the first analysis, the indirect comparisons among pharmacological agents clearly favored the combination of isatuximab plus dexamethasone (Figure 4A); in the analysis on cell-based therapies (Figure 4B), a significantly better OS was found for ciltacabtagene autoleucel compared with idecabtagene vicleucel (HR, 0.47; 95% CI, 0.29 to 0.75, *p* = 0.0014). The investigation by Messori et al. [6] was mainly aimed at estimating the gain in PFS determined by idecabtagene vicleucel (Figure 5). Overall, the Shiny method made it possible to compare numerous innovative products in the absence of direct comparisons.

### 2.4. CAR T-Cells vs. Pharmacological Agents in Non-Hodgkin Lymphoma (NHL)

Numerous Shiny method investigations have focused on NHL, especially relapsed-refractory diffuse large B-cell lymphoma. The paper by Messori et al. [7] analyzed the most updated information about the survival pattern determined by the two CAR T-cell products currently available for this clinical indication (i.e., axicabtagene ciloleucel, also known as axi-cel, and tisagenlecleucel, or tisa-cel). Axi-cel seems to improve OS to a greater extent than tisa-cel (Figure 6). On the other hand, the effectiveness of pharmacological agents has improved in the past five years. As a result, the therapeutic scenario has changed because tisa-cel seems now to offer less interest for these patients whereas pharmacological agents are becoming more and more effective. 

### 2.5. Pharmacological Agents in Relapsed-Refractory NHL

The Shiny method analysis by Messori and Caccese [8] compared four pharmacological agents for relapsed-refractory diffuse large B-cell lymphoma. Its results underscore a remarkable efficacy of tafasitamab plus lenalidomide compared with polatuzumab vedotin, loncastuximabtesirine, and selinexor (Figure 7). Considering the extent of this difference in favor of tafasitamab plus lenalidomide, the question arose on whether the patients of that trial were prognostically favored compared with those enrolled in the other three trials. More recently, this analysis involving four agents was updated by adding glofitamab [9], a bispecific antibody, as a fifth comparator. In this extended analysis (Figure 8), glofitamab failed to demonstrate any advantage over the other four comparators; in this case, the Shiny method contributed to identify a worse OS profile for this bispecific antibody than that initially expected.

### 2.6. First-Line Treatments in Advanced Melanoma

In 2022, a Shiny method analysis [10] investigated the emerging combination of immune checkpoint inhibitors (i.e., relatlimab plus nivolumab) compared with the previously available immunotherapies for advanced or metastatic melanoma. Six treatments were compared: pembrolizumab plus ipilimumab, nivolumab plus ipilimumab (two dosages), relatlimab plus nivolumab, pembrolizumab monotherapy, and nivolumab monotherapy (Figure 9A). In this case, in the original trial conducted by Tawbi et al., the novel combination of relatlimab plus nivolumab was superior compared with nivolumab monotherapy. However, monotherapy with immune checkpoint inhibitors, either pembrolizumab or nivolumab, is no longer considered the standard of care because the combination of ipilimumab + nivolumab has replaced the monotherapy. Therefore, this Shiny method analysis permitted indirect comparison between relatlimab plus nivolumab and the real standard of care (ipilimumab plus nivolumab; Figure 9A). The results indicate that relatlimab plus nivolumab shows only a trend towards better PFS compared to nivolumab plus ipilimumab while pembrolizumab plus ipilimumab seems to be the most effective first-line therapy, even though this result requires further validation in a double-blind controlled study.

This example is interesting because it shows that the easy indirect comparisons generated by the Shiny method can help to correctly interpret the survival data and permit the standard of care used for comparison to be kept updated. On the other hand, the superiority of relatlimab plus nivolumab vs. nivolumab monotherapy in Tawbi’s trial can be explained because the controls of Tawbi’s trial had more favorable prognostic characteristics than the controls of Wolchok’s trial that proposed nivolumab as standard of care (Figure 9B).

### 2.7. Second-Generation Hormone Treatments (Enzalutamide, Apalutamide, and Daralutamide) Proposed for Nonmetastatic Castration-Resistant Prostate Cancer (M0CRPC)

In a Shiny method analysis [11], the efficacy of second-generation hormone treatments (enzalutamide, apalutamide, and daralutamide) proposed for M0CRPC was investigated because there were no direct comparisons between these agents. Following the publication of updated results regarding the OS of these treatments, information was summarized according to a Shiny method graph (Figure 10A). The objective was to assess the degree of equivalence among these treatments. The values of HR vs. placebo from the reconstructed OS curves were statistically significant for each of the three comparisons. Apalutamide, darolutamide, and enzalutamide showed better efficacy in comparison, respectively, with apalutamide (HR = 0.75; 95% confidence interval [CI], 0.64 to 0.88), darolutamide (HR = 0.70; 95% CI, 0.58 to 0.84); and enzalutamide (HR = 0.77; 95% CI, 0.65 to 0.90). These results showed no difference in OS between any of these three agents and, in testing for equivalence, these HRs met the criterion at the 0.75 to 0.33 level (Figure 10A), i.e., around ±25%. On the other hand, there was no heterogeneity in this analysis (Figure 10B).

### 2.8. Treatments for Metastatic Urothelial Cancer

This study [12] represents the first experience of the so-called “one-to-many” comparative approach. Several (or “many”) treatments were available for advanced or metastatic urothelial carcinoma, mainly consisting of immunotherapy or chemotherapy, and they had been previously analyzed by Rivano et al. [13] according to the restricted mean survival time (RMST). At the time of the Shiny method analysis, a new potentially innovative treatment (“one”) was proposed (enfortumab vedotin) in the light of a phase-II trial. Our “one-to-many” analysis (Figure 11A,B) aimed to assess the benefits of the new treatment in comparison with the alternatives developed previously. Albeit enfortumab vedotin does not share the same line of treatment, this new treatment showed a better OS than immune checkpoint inhibitors (ICIs). On the other hand, standard chemotherapy and/or vinflunine were less effective (Figure 11A,B). The heterogeneity assessment evaluating the control arms revealed that the survival gain produced by enfortumab vedotin might be, to some extent, influenced by the better prognosis of the population enrolled in the enfortumab trial in comparison with patients enrolled in the three ICI trials.

### 2.9. First-Line Therapy for Metastatic Triple-Negative Breast Cancer

The Shiny method analysis on first-line immunotherapies in triple-negative in breast cancer [14] was published in January 2022, but its results were appropriately interpreted later on [15]. Apparently, in the indirect comparison (endpoint, OS), atezolizumab seems to be significantly more effective than pembrolizumab (Figure 12A) but looking at the heterogeneity of the control groups modifies this interpretation. In fact, the controls of the atezolizumab trial, treated mainly with taxane-based chemotherapy, had a significantly better OS than those of the pembrolizumab trial, treated with either taxane or platinum + gemcitabine (Figure 12B). Hence, the better OS of atezolizumab in the indirect comparison with pembrolizumab may depend on the characteristics of the atezolizumab’s controls rather than on a better pharmacological activity of atezolizumab. Possible explanations are that taxane-associated chemotherapy is more effective than platinum plus gemcitabine; alternatively, patients enrolled in the atezolizumab trial might have more favorable-baseline characteristics, as suggested by the better OS of its controls. 

### 2.10. Maintenance Therapy Using Poly-ADP-Ribose Polymerase Inhibitors (PARPIs) as a New Therapeutic Option in Advanced Ovarian Cancer

Maintenance therapies for patients with advanced ovarian cancer who responded to platinum therapy were examined by Cancanelli et al. [16] with an analysis based on RMST. Cancanelli et al. compared olaparib, niraparib, and rucaparib based on the PFS of five placebo-controlled studies. The conclusion was that these therapies are more effective than placebo with gains in RMST of around 6 to 8 months. In another more recent analysis based on the Shiny method [17], each of these three maintenance treatments showed a significant improvement in PFS compared with the controls (Figure 13). Interestingly enough, in comparing olaparib with niraparib or rucaparib, the extent of this improvement was significantly greater with olaparib. One confounding factor in these indirect comparisons across these three PARPIs is that, of the two studies assessing olaparib, the one published by Moore et al. selectively enrolled patients with BRCA-mutation as opposed to those without BRCA mutation or with unknown BRCA status. Hence, the superior PFS and RMST values with olaparib compared to niraparib or rucaparib may be explained considering that—in this specific type of cancer—BRCA-mutation determines a more favorable prognosis.

### 2.11. Cryoballon and Radiofrequency Ablation in Paroxysmal Atrial Fibrillation

This analysis illustrates how the Shiny method can be applied in settings other than haemato-oncology. The Shiny method analysis by Trippoli et al. [18] re-examined a therapeutic issue previously studied in a network meta-analysis based on RMST [19]. In the Shiny method analysis [18], the clinical evidence published for cryoballoon ablation (CBA), radiofrequency ablation (RFA), and medical therapy in patients with paroxysmal atrial fibrillation was assessed according to the endpoint of time to recurrence of atrial fibrillation (or other forms of atrial arrhythmias). The results based on reconstructed patients show that CBA yields a significant improvement compared to the controls (Figure 14), whereas RFA fares slightly better than controls but not at levels of statistical significance. In comparison with medical therapy, the improvement estimated through the HR is greater for CBA than for RFA. The indirect comparison between CBA versus RFA favors the former at levels of statistical significance. If one compares the results provided by the Shiny method [18] with those published in the previous RMST meta-analysis [19], this comparison provides interesting insights into the advantages and disadvantages of these two approaches (see Reference [18] text for details).

### 2.12. Heterogeneity as a Key Factor Influencing Efficacy

As previously discussed, one of the main critical aspects of the Shiny method is represented by a biased selection of the studies included in the analysis. Hence, in all Shiny method analyses it is essential to carefully evaluate whether the criteria used to enroll the patients were homogenous among the trials. Despite this, trials that apparently enrolled patients with similar inclusion/exclusion criteria can be affected all the same by a certain degree of heterogeneity in patient selection, and this unnoticed heterogeneity can impact efficacy. To address this problem, a heterogeneity assessment on control groups is suggested for all Shiny analyses [15]. In fact, if patients’ characteristics are similar at baseline, the control arms of different trials are expected to behave homogenously in terms of survival. This can be demonstrated by performing the analyses on control groups previously described in Figure 10B (for prostate cancer [11]) and Figure 12B (for triple negative breast cancer [14]). The likelihood ratio test [15] is suitable for these analyses, but other parameters are currently being investigated. The likelihood ratio test can be used only within parametric models.

To discuss this issue more extensively, two data sets [11,14] deserve to be examined in more detail: (a), The comparison of treatments for nonmetastatic castration-resistant prostate cancer (3 trials with 3 active treatments—apalutamide, darolutamide, and enzalutamide—and 3 control groups given placebo) [11]; (b), The comparison of treatments for triple-negative breast cancer (3 trials, with 2 active treatments—i.e., atezolizumab and pembrolizumab—and 3 control groups given chemotherapy plus placebo) [14].

In the first case (Shiny method analysis on prostate cancer [11]), the degree of heterogeneity is low (Figure 10B), thus suggesting full reliability of the indirect comparisons that demonstrated an overall equivalence of these drugs. In the second case (Shiny method analysis on triple-negative breast cancer [14]), the better efficacy of atezolizumab over pembrolizumab is a conclusion that needs to be reconsidered because the heterogeneity assessment reveals that controls of atezolizumab trials performed significantly better than those of pembrolizumab trials (Figure 12B); this may be due to differences in the chemotherapy regimen associated with immune checkpoint-inhibitors.

### 2.13. Therapeutic Equivalence

Demonstrating therapeutic equivalence among medicines is the last application of the Shiny method [20] discussed in the present article. In Italy, the National Agency of Medicines (AIFA) is the Institution that has responsibility to grant nation-wide permission to run competitive tenders in the area of medicines, namely in the procurement of medicine to be used in public hospitals. To obtain this permission, Regions must submit to AIFA an evidence-based report focused on the pharmacological class demonstrating that between-drug variabilities in the main efficacy endpoint are kept within the statistical limit of ±10%. If the available evidence supports this demonstration, competitive in-hospital tenders can be run to purchase the medicines belonging to the class studied by the report. At least two Shiny method analyses have already been aimed at this objective: the first [17] has been carried out to pursue the equivalence among olaparib, rucaparib, and niraparib as maintenance treatment in patients responding to platinum therapy in ovarian cancer; PFS was the clinical endpoint. In this case, the demonstration of equivalence failed because olaparib proved to be more effective than the other two PARP inhibitors (Figure 14). Another example in the demonstration of equivalence regards atezolizumab and pembrolizumab [20] given as first-line immunotherapy in patients with non-small cell lung cancer and PD(L) expression >50%. (Figure 15A); in this case, the demonstration of equivalence was successful owing to the low degree of heterogeneity (Figure 15B).

### 2.14. Expanding Experiences Worldwide

In recent years, different researchers worldwide have recognized the scientific validity of the Shiny method [21] and have applied it both in systematic reviews and meta-analyses, mostly regarding haemato-oncological drugs. Some applications have been published in the field of medical devices, e.g., to provide a pooled estimate of the risk of lead failure in implantable cardioverter-defibrillators [22] and to investigate the prognosis and persistence of smell and taste dysfunction in patients with COVID-19 [23].

Although no large-scale studies have specifically investigated the precision of the method, anecdotical information agrees that the method provides excellent reconstructions of the Kaplan-Meier curves. On the basis of the trials on CLL examined in Figure 3, we compared the values of HR reported in the original trials with those estimated from the curves reconstructed by the Shiny method (Figure 16). Four trials were suitable for this comparison; even though the analysis was purely descriptive, also these results (Figure 16) support the conclusion that the Shiny method has a notable precision. Other studies focusing on the precision of the Shiny method are ongoing (Damuzzo et al. unpublished observations), and their results are expected in the next months.

## 3. Conclusions and Future Directions

The reconstruction of individual patient data (IPD) to perform comparative studies is an innovative area of clinical research in which indirect comparisons represent the most typical objective. The 14 experiences reviewed in this article indicate that this type of research is nowadays supported by a sound methodology. However, one should keep in mind that these indirect comparisons, which by definition have a retrospective design, are unable to account for some “between-trial” differences in enrolled patients that can have an important impact on outcomes. To minimize this bias, before conducting a Shiny method analysis, a careful assessment is needed about the inclusion criteria adopted in the different trials. Furthermore, after finishing a Shiny method analysis, post-hoc statistical assessments are always needed to explore the degree of heterogeneity affecting control groups of included trials. Further and more specific tests still need to be developed to improve the performance of these post-hoc assessments of heterogeneity. All in all, this review confirms that the main strength of the Shiny method approach lies in the communication strength of its typical multi-treatment graphs, which represent a new and easily interpretable tool in the armamentarium of evidence-based medicine.

## Figures and Tables

**Figure 1 cancers-15-01633-f001:**
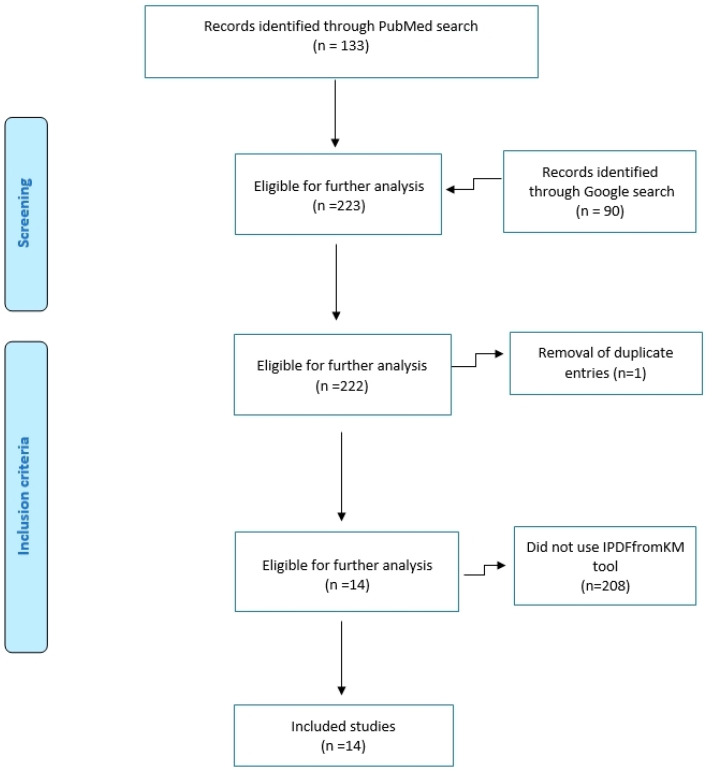
PRISMA schematic of our literature search.

**Figure 2 cancers-15-01633-f002:**
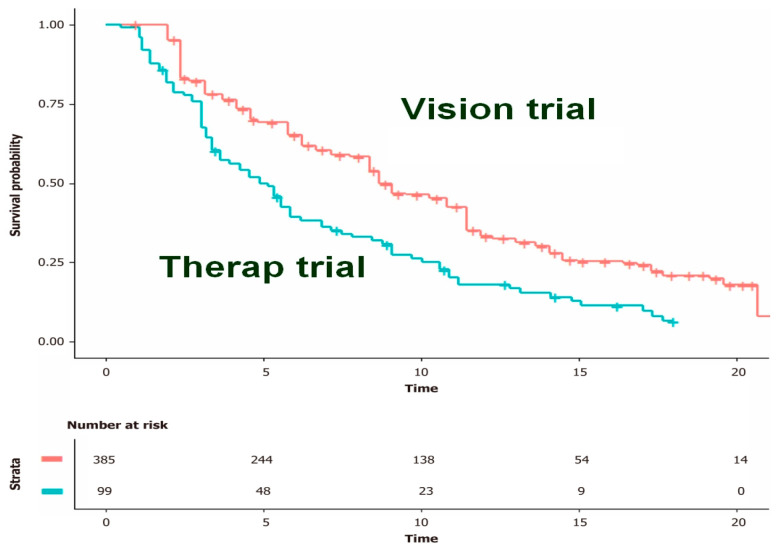
Kaplan-Meier curves from reconstructed patient-level data. Pooled Kaplan-Meier survival curves obtained by reconstruction of IPD from two trials (therapy and vision). Vision trial in red, therap trial in blue; time expressed in months. Modified from reference [1].

**Figure 3 cancers-15-01633-f003:**
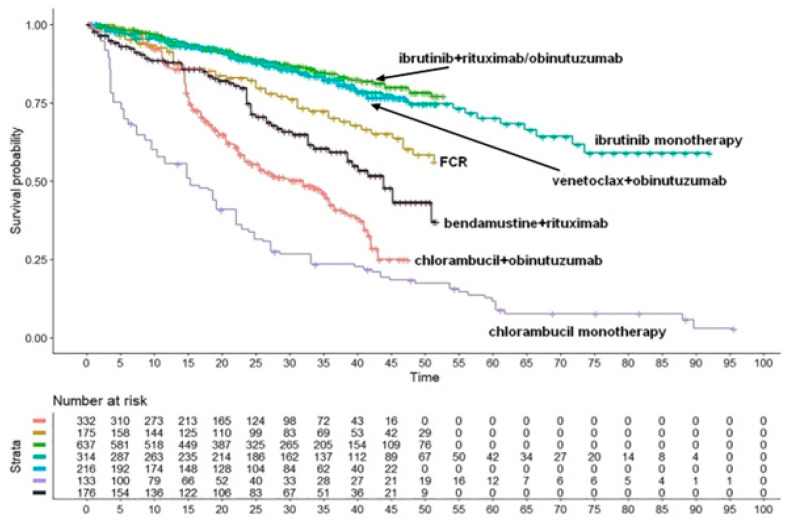
Chronic lymphocytic leukemia. Survival curves from the reconstruction of IPD from 5 trials. Ibrutinib+rituximab/obinutuzumab (637 patients; green curve), ibrutinib monotherapy (314 patients; light green curve), venetoclax + obinutuzumab (216 patients; light blue curve), FCR (175 patients; gold curve), bendamustine + rituximab (176 patients; black curve), chlorambucile + obinutuzumab (332 patients; orange curve) and chlorambucile monotherapy (133 patients; violet curve). Endpoint, progression-free survival. Time expressed in months. Abbreviations: FCR, fludarabine+cyclophosphamide+rituximab. Long-term progression-free survival curves modified from reference [4].

**Figure 4 cancers-15-01633-f004:**
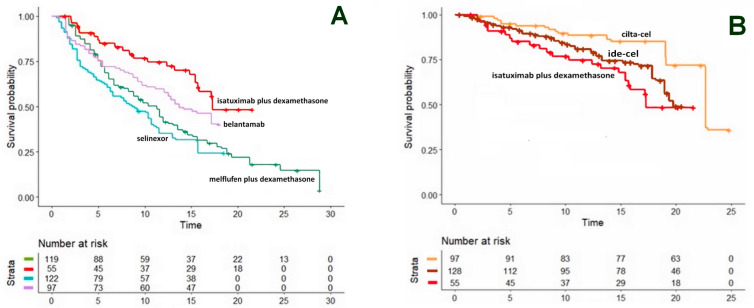
Relapsed-refractory multiple myeloma. (**A**). OS curves from the reconstruction of IPD from 4 trials. Isatuximab + dexamethasone (NCT01084252 trial; 55 patients, red curve), belantamab monotherapy (DREAMM-2 trial; 97 patients, violet curve), melflufen + dexamethasone (HORIZON trial; 119 patients, green curve) and selinexor monotherapy (STORM trial; 122 patients, light blue curve). (**B**) OS curves from the reconstruction of IPD from 3 trials, ciltacabtagene autoleucel (CARTITUDE-1 trial; 97 patients, orange curve), idecabtagene vicleucel (KarMMa trial; 128 patients, brown curve) and isatuximab plus desamethasone (NCT01084252 trial; 55 patients, red curve). Endpoint, overall survival. Time expressed in months. Overall survival curves modified from reference [5].

**Figure 5 cancers-15-01633-f005:**
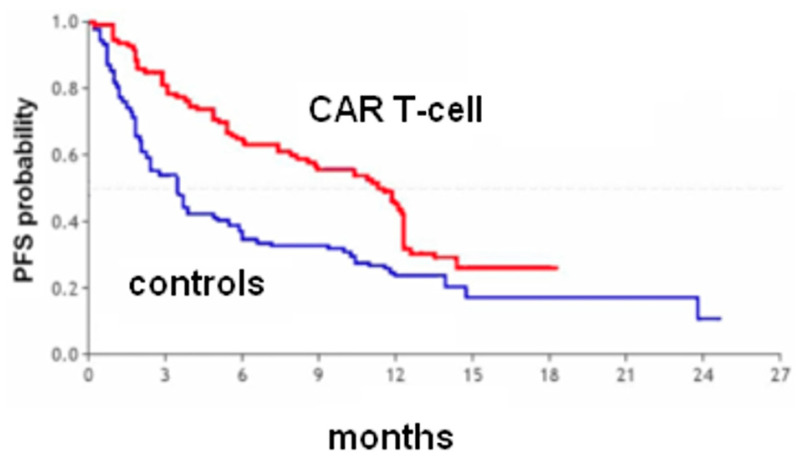
Heavily pretreated multiple myeloma. PFS curves from the reconstruction of IPD from 1 trial [Jagannath et al. 2020]. Idecabtagene vicleucel (128 patients, red curve) and matched controls (190 patients, blue curve). Endpoint, progression-free survival. Time expressed in months. Progression-free survival curves modified from reference [6].

**Figure 6 cancers-15-01633-f006:**
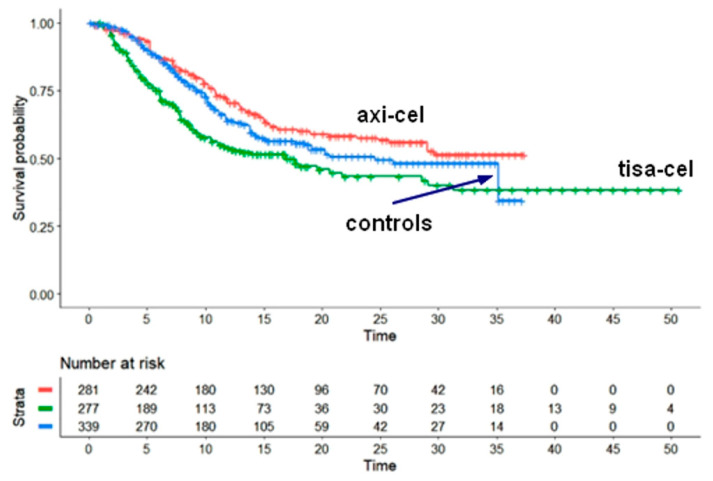
Large B-cell lymphoma. Survival curves from the reconstruction of IPD from 4 trials. Axicabtagene ciloleucel (pooled survival from two trials (ZUMA-1 and ZUMA-7 trials; 281 patients, red curve), Tisagen lecleucel (pooled survival from 2 trials (JULIET and BELINDA trials; 277 patients, green curve) and pooled control group from 2 trials (ZUMA-7 and BELINDA trials; 339 patients, blue curve). Endpoint, overall survival. Time expressed in months. Overall survival curves modified from reference [7].

**Figure 7 cancers-15-01633-f007:**
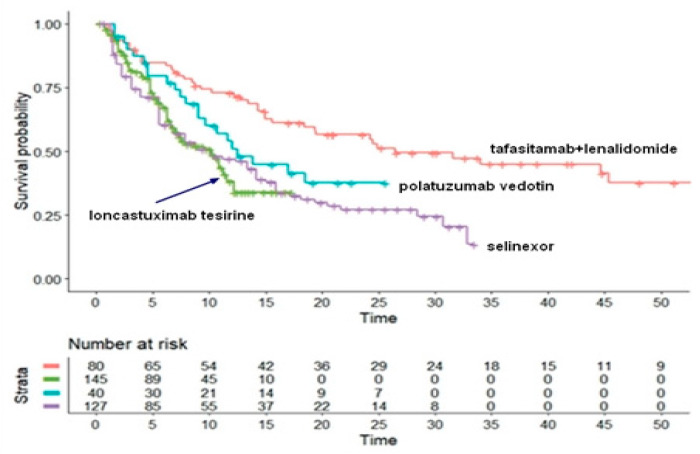
Relapsed-refractory diffuse large B-cell lymphoma. Survival curves from the reconstruction of IPD from 4 trials. Tafasitamab + lenalidomide (80 patients, orange curve), polatuzumab vedotin (40 patients, light blue curve), selinexor (127 patients, violet curve), and loncastuximab tesirine (145 patients, green curve). Endpoint, overall survival. Time expressed in months. Overall survival curves modified from reference [8].

**Figure 8 cancers-15-01633-f008:**
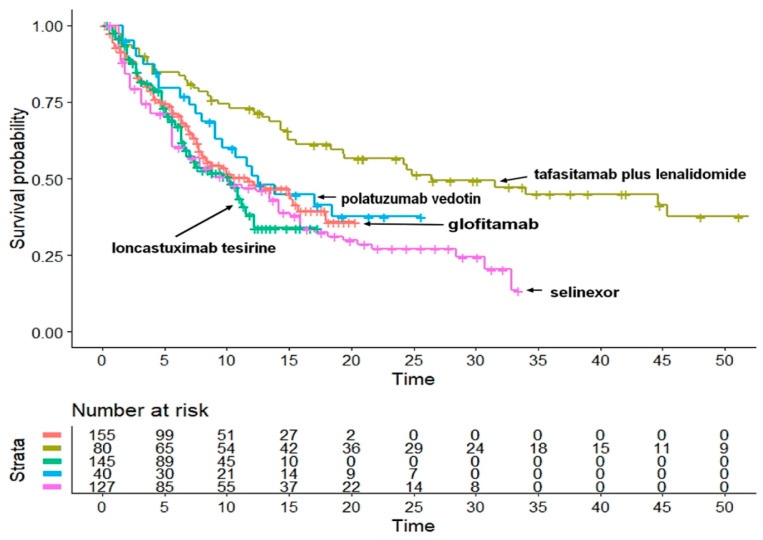
Relapsed-refractory diffuse large B-cell lymphoma. Survival curves from the reconstruction of IPD from 5 trials. Tafasitamab + lenalidomide (dark green curve), polatuzumab vedotin (light blue curve), selinexor (violet curve), loncastuximab tesirine (green curve), and glofitamab (red curve). Endpoint, overall survival. Time expressed in months. Overall survival curves modified from reference [9].

**Figure 9 cancers-15-01633-f009:**
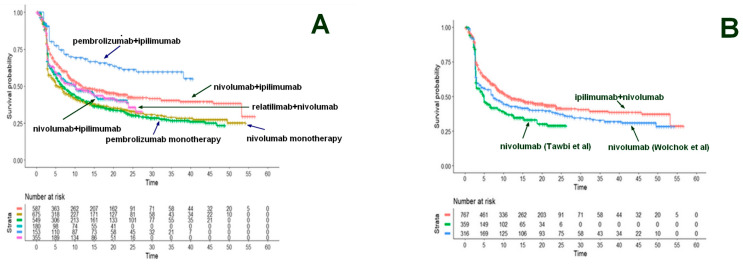
Advanced melanoma. (**A**). Survival curves from the reconstruction of IPD from 6 trials. Nivolumab 1 mg/kg + ipilimumab 3 mg/kg (from Checkmate-067, Checkmate-511, and Checkmate-069 trials; 587 patients, red curve); nivolumab monotherapy (from RELATIVITY- 047 and Checkmate-067 trials; 675 patients, dark green curve); pembrolizumab monotherapy (from Keynote-006 trial; 549 patients, light green curve); nivolumab 3 mg/kg + ipilimumab 1 mg/kg (from Checkmate-511 trial; 180 patients, light blue curve); pembrolizumab + ipilimumab (from Keynote-029 trial; 153 patients, dark blue curve) and relatlimab + nivolumab (from RELATIVITY-047 trial; 355 patients; purple curve). (**B**) Survival curves from the reconstruction of IPD from 4 trials. Ipilimumab + nivolumab at various dosages (from Wolchok et al., 2022, Lebbé et al., 2019, and Hodi et al., 2016; 767 patients, red curve); nivolumab monotherapy (from Tawbi et al., 2022; 359 patients, green curve) and nivolumab monotherapy (from Wolchok et al., 2022; 316 patients, light green curve); nivolumab 3 mg/kg + ipilimumab 1 mg/kg (from Checkmate-511 trial; 180 patients, blue curve). Endpoint, progression-free survival. Time expressed in months. Progression-free survival curves modified from reference [10].

**Figure 10 cancers-15-01633-f010:**
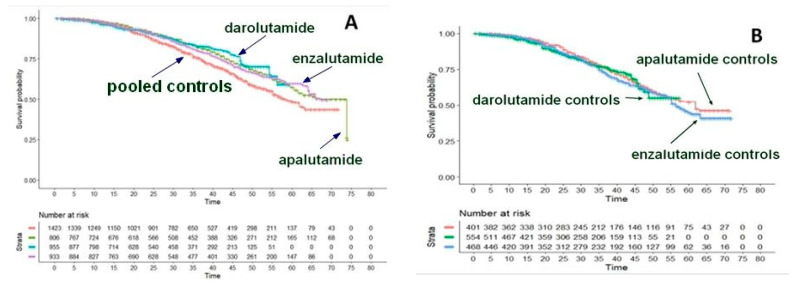
Nonmetastatic castration-resistant prostate cancer. (**A**). Comparison of survival curves from the reconstruction of IPD from the three active treatment cohorts. Darolutamide (ARAMIS trial; 955 patients, blue curve), enzalutamide (PROSPER trial; 933 patients, purple curve), apalutamide (SPARTAN trial; 806 patients, green curve), and pooled controls (1423 patients, red curve). (**B**). Comparison of survival curves from the reconstruction of IPD from the three placebo cohorts. Darolutamide (ARAMIS trial; 554 patients, green curve), enzalutamide (PROSPER trial; 468 patients, blue curve), apalutamide (SPARTAN trial; 401 patients, red curve). Endpoint, overall survival. Time expressed in months. Overall survival curves modified from reference [11].

**Figure 11 cancers-15-01633-f011:**
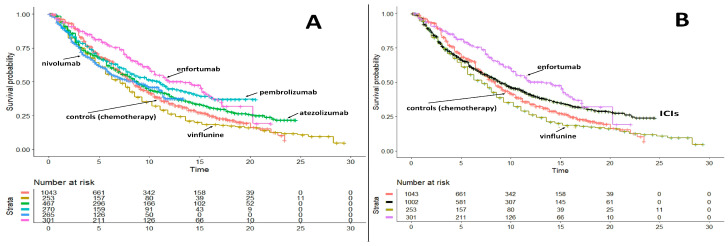
Advanced or Metastatic Urothelial Carcinoma. (**A**) Comparison of survival curves from the reconstruction of IPD in “one-to-many” survival analysis. Enfortumab (301 patients, purple curve), nivolummab (265 patients, light blue curve), pembrolizumab (270 patients, light green curve), atezolizumab (467 patients, green curve), vinflunine (253 patients, gold curve), pooled controls (1043 patients, red curve). (**B**) Same data reported in panel A but immunotherapies were pooled into a single group. Endpoint, overall survival. Time expressed in months. Overall survival curves modified from reference [12].

**Figure 12 cancers-15-01633-f012:**
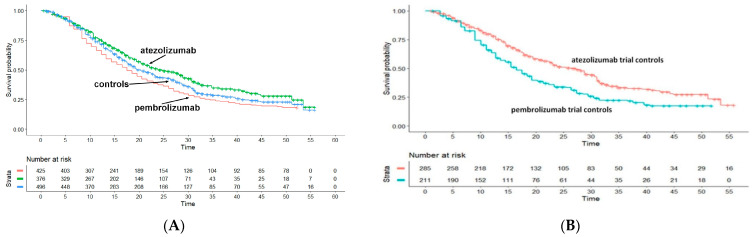
Triple-Negative Breast Cancer. Survival curves from the reconstruction of IPD from 3 trials. Pembrolizmab (KEYNOTE-355 trial; 425 patients, red curve), atezolizumab (IM-PASSION-130 and IM-PASSION-131 trial; 376 patients, green curve) and pooled controls of the three trials (496 patients, blue curve). Endpoint, overall survival. Time expressed in months. Overall survival curves modified from references [14,15].

**Figure 13 cancers-15-01633-f013:**
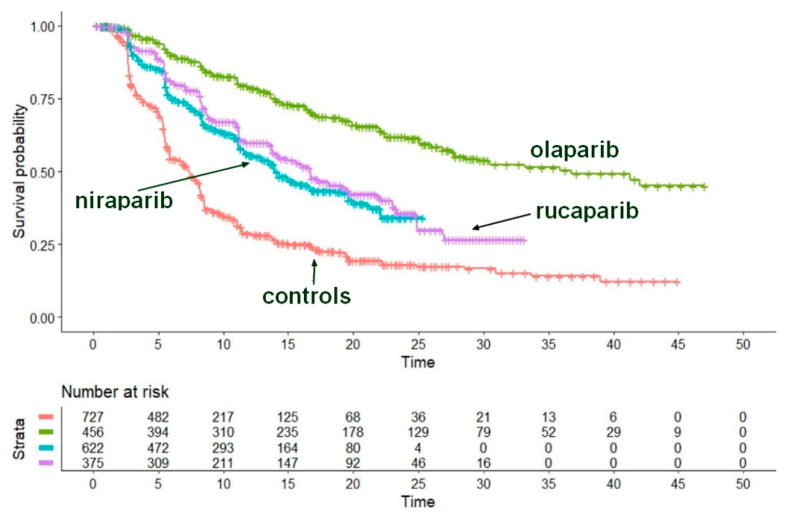
Ovarian carcinoma. Survival curves from the reconstruction of IPD from 5 trials. Olaparib (Pujade-Lauraine et al., 2017 and Moore et al., 2018; 456 patients; green curve), rucaparib (Coleman et al., 2017; 375 patients; purple curve), niraparib (Mirza et al., 2016 and González-Martín et al., 2019; 622 patients; light blue curve), and pooled controls (727 patients; red curve). Endpoint, progression-free survival. Time expressed in months. PFS curves modified from reference [17].

**Figure 14 cancers-15-01633-f014:**
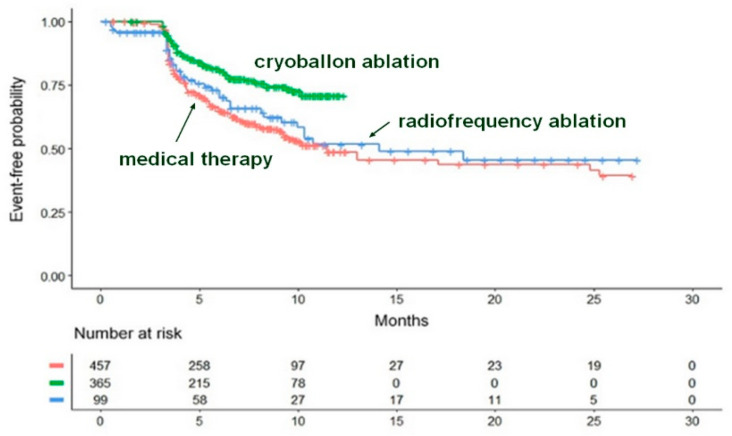
The three curves refer to cryoballoon ablation, radiofrequency ablation, and medical therapy. Recurrence of atrial fibrillation or of another atrial arrhythmia is the time-to-event endpoint. Identification of the curves: in green, cryoballoon ablation (three trials, n = 365); in blue, radiofrequency ablation (two trials, n = 99); in red, medical therapy (five trials, n = 457). Modified from reference [18].

**Figure 15 cancers-15-01633-f015:**
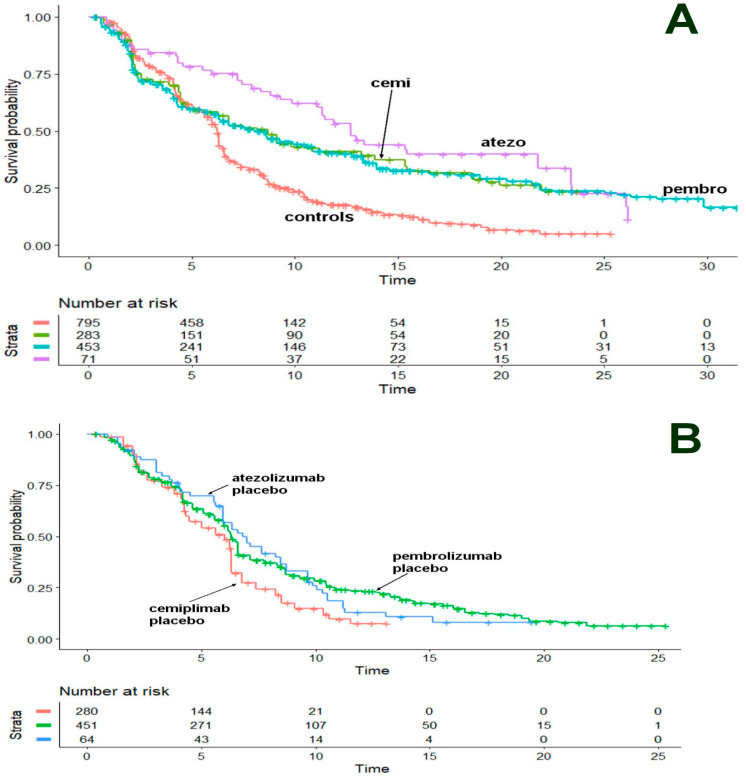
First-line immunotherapy in patients with non-small cell lung cancer and PD(L) expression >50%. Panel (**A**): progression-free survival in the three experimental patient groups; three placebo control groups; the fourth curve refers to all control groups pooled together. Panel (**B**): heterogeneity assessment in the three control groups. Endpoint, progression-free survival, in months. Modified from reference [20].

**Figure 16 cancers-15-01633-f016:**
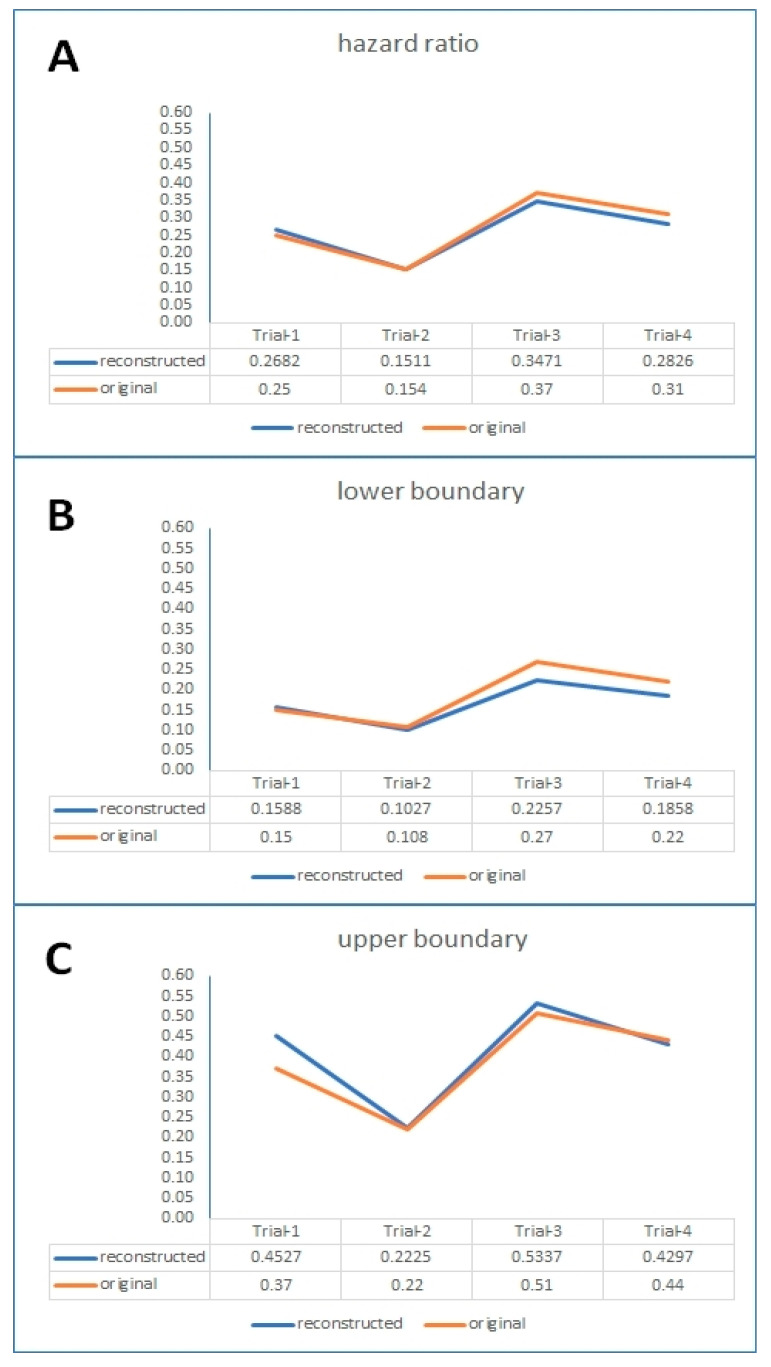
Descriptive comparison of values of hazard ratio (along with 95% lower boundary and 95% upper boundary; Panels (**A**), (**B**), (**C**), respectively) reported in the original trials with the values estimated from reconstructed Kaplan-Meier curves. Data regarding treatments for CLL studied in 4 trials (Trial-1, ILLUMINATE: I + O vs. C + O; Trial-2, RESONATE-2: I vs. C; Trial-3, NCT02048813: I + R vs. FCR; Trial-4, NCT02242942: V + O vs. C + O; see reference [4]). Abbreviations: I, ibrutinib; O, obinutuzumab; C, chlorambucil; R, rituximab; FCR, fludarabine + cyclophosphamide + rituximab.; V, venetoclax.

## Data Availability

The data presented in this study are available on request from the corresponding author.

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
