# Peer review of "Application of the IPDfromKM-Shiny Method to Compare the Efficacy of Novel Treatments Aimed at the Same Disease Condition: A Report of 14 Analyses"

_cancers, 2023, doi:10.3390/cancers15061633_

Round 1

Reviewer 1 Report

I read a review manuscript entitled “Application of the Shiny method to compare the efficacy of novel treatments aimed at the same disease condition: a report of 14 analyses” written by Andrea Messori. Actually, I’ve not been to hear about “Shiny Method” which looks advanced new optimal statistics to compare the trial differences by abrogating inter-trial variations.

Practical analysis of 14 cases’ presentation to explore an instance of this statistical method were helpful to understand and review the Shiny method. However, I’m not a biostatistician. Then, I would like to offer providing an explanatory comment in the section “2.12. Heterogeneity as a key factor influencing efficacy,” for non-specialist readers. How was the methodology for treating heterogeneity to assure comparability?

What is the conceptual procedure or algorithm to qualify to compare heterogeneity, such as propensity scores, adjusting standard deviation, principal component analysis, clustering method, positioning mapping or something like that?

Reviewer 2 Report

I am happy to read your interesting work discussing the usefulness of Shiny method to comparison with drug efficacy, and here are my recommend for you.

1. Would you mind sharing your Shiny codes used for the analyses in your work to allow many colleagues to perform and re-evaluate and feed back. please?

2. I think the summary of comparesion the shiny with conventional KM and  HR analyses would be added in short in your conclusion section. 

Reviewer 3 Report

In the submitted manuscript, the authors report using a new method (the “Shiny method”) to generate individual patient data from the published Kaplan-Meier (KM) curves and perform indirect comparisons. They concluded that the “Shiny method” proves to be a valid advancement for conducting comparative research in the area of evidence-based medicine.  Although this is in interesting manuscript, there are a number of deficiencies noted.

1.      To be more precise, the method should be called “IPDfromKM using the Shiny application” or “IPDfromKM/Shiny”.  Shiny is an R package that makes it easy to build interactive web apps straight from R. It is not a “method” per say.  Shiny can be used in a wide variety of applications and is a generic name which does not reflect what “method” uses to generate individual patient data. As the authors noted, the “IPDfromKM” method has two renditions: an R package and a Shiny app. For abbreviation, it is better just called “IPDfromKM” rather than “Shiny”.

2.      The authors report the result of 14 analyses applying “IPDfromKM/Shiny”.  In addition to what’s reported, it will be desirable to add the point estimates and its confidence intervals of hazard ratio, RMST and show the results in forest plots in some of the 14 analyses. For example, in Figure 3, one can construct HRs of each other treatments using the chlorambucile monotherapy as a reference group.

3.      The authors state: “Shiny method” is midway between artificial intelligence and advanced statistics. (sic).  This is an overstatement.  One may say that it is midway between simple interpolation and advanced statistics.     

4.      To describe the method, it requires either total number of patients and/or the number of patients at risk by time intervals.  It is important to note that adding the number of patients at risk by time intervals can increase the precision of the generation of individual patient data.

5.      In many places, it is more accurate to state that the individual patient data are “generated” or “reconstructed” rather than “simulated”.  There is no simulation performed.  Although simulation can be performed after fitting the KM curves with parametric models or using bootstrap to resample the generated individual patient data, I don’t think the authors performed simulation in this manuscript.

6.      Figure 2 and other similar figures: How can one tell that the method does a good job?  In some examples, it will be useful to show the original and the reconstructed curves in two side-by-side panels for comparison. One may compare the number of events and censoring of the original data and the generated data, if possible.

7.      Page 3, Lines 111-112: “metastatic castration-resistant prostate “cancer”.

8.      Page 5, Lines 165-166: By convention, a hazard rate of the reference group should be in the denominator when computing the HR.  Thus, HR<1 indicates that the experimental group is better than the control group.  In this example, HR > 1 but the author claimed that a significantly better OS for ciltacabtagene autoleucel compared with idecabtagene vicleucel.”  Also, from the figure, should it be “idecabtagene vicleucel”?

9.      Figure 4B: The color schemes of the figure and legends do not match. Also, should the treatment be “axicabtagene ciloleucel”?

10.   Please add the caveat of the potential “time drift” when pooling or comparing trials at different time period.

11.   One benefit of applying IPDfromKM is that, by applying the arm-based analysis of the network meta-analysis, arms in different trials can be compared.

12.   Page 9, Line 282: Should .33 be 0.33?

13.   Page 9, Lines 306-307: How can one tell simply from KM curves that some groups have better prognostic factors?

14.   Page 12, Line 394: Please note that likelihood ratio test can only be applied after fitting the data with parametric models.

Round 2

Reviewer 3 Report

Please add "IPDfromKM" in front of the "Shiny Method" in the title, the abstract, and the body of the manuscript.  Using the full name "IPDfromKM Shiny Method" is important.  After first appearing, you may abbreviate to "Shiny Method."

Author Response

We have revised our manuscript according to the suggestion made by Reviewer 3 (please see the attachment).  In the revised version, we have removed the quotes from the method name; if deemed preferable, we can still keep the quotes.
